# Pain-Pressure Threshold Changes throughout Repeated Assessments with No Sex Related Differences

**DOI:** 10.3390/healthcare11040475

**Published:** 2023-02-07

**Authors:** Andreas Konrad, Kazuki Kasahara, Riku Yoshida, Yuta Murakami, Ryoma Koizumi, Masatoshi Nakamura

**Affiliations:** 1Institute of Human Movement Science, Sport and Health, Graz University, Mozartgasse 14, 8010 Graz, Austria; 2Institute for Human Movement and Medical Sciences, Niigata University of Health and Welfare, 1398 Shimamicho, Kitaku, Niigata 950-3198, Japan; 3Department of Physical Therapy, Niigata University of Health and Welfare, 1398 Shimamicho, Kitaku, Niigata 950-3198, Japan; 4Faculty of Rehabilitation Sciences, Nishi Kyushu University, 4490-9 Ozaki, Kanzaki 842-8585, Japan

**Keywords:** pain-pressure threshold, algometer, muscle, pain

## Abstract

Algometers are commonly used to measure the pain-pressure threshold (PPT) in various tissues, such as muscle, tendons, or fascia. However, to date, it is not clear if the repeated application of a PPT assessment can adjust the pain thresholds of the various muscles. Therefore, the purpose of this study was to investigate the repeated application of PPT tests (20 times) in the elbow flexor, knee extensor, and ankle plantar flexor muscles in both sexes. In total, 30 volunteers (15 females, 15 males) were tested for their PPT using an algometer on the respective muscles in random order. We found no significant difference in the PPT between the sexes. Moreover, there was an increase in the PPT in the elbow flexors and knee extensors, starting with the eighth and ninth assessments (out of 20), respectively, compared to the second assessment. Additionally, there was a tendency to change between the first assessment and all the other assessments. In addition, there was no clinically relevant change for the ankle plantar flexor muscles. Consequently, we can recommend that between two and a maximum of seven PPT assessments should be applied so as not to overestimate the PPT. This is important information for further studies, as well as for clinical applications.

## 1. Introduction

Algometers are typically used to measure the pain-pressure threshold (PPT) in various tissues, such as muscles [1], tendons [2], and fascia [3]. In general, during a PPT assessment, the metal rod of the algometer is attached to the tissue of interest by continuously increasing the pressure applied by the therapist/researcher. The patient/participant is asked to give verbal feedback to the therapist/researcher when pain occurs, rather than just pressure [4]. Since the location of the target tissue can be easily highlighted on the skin, the reliability of such a measurement has been reported to be good to excellent [5,6], especially when the same therapist/researcher has performed the measurement [5]. With regard to exclusively muscle, previous studies have investigated the effects of flexibility-enhancing stimuli such as stretching and foam rolling on the PPT [4,7,8]. Most studies have reported an increase in PPT following such interventions, and hence it was concluded that increased pain tolerance (i.e., stretch tolerance) was the main variable responsible for the increase in flexibility (i.e., range of motion). In addition, various studies have conducted PPT assessments to estimate muscle soreness after a workout [9,10].

However, it should be mentioned that the previous published studies used various protocols when using the algometer. While some studies applied only one PPT assessment [8,11], others, however, used multiple PPT assessments [4,12]. Since pain is induced during the PPT measurement, it can be assumed that multiple assessments with an algometer can alter pain sensitivity, due to the repeated application. Consequently, by taking the average or the highest value of PPT, overestimation can occur. Therefore, the goal of this study was to perform multiple PPT assessments (i.e., 20) in three different muscles of the dominant side (elbow flexor, knee extensor, and ankle plantar flexor muscles) to detect if there is a shift in the PPT throughout the repeated PPT assessments. Moreover, a further goal was to detect differences between the sexes in the potential change in the PPT throughout the repeated assessments.

## 2. Materials and Methods

### 2.1. Experimental Approach to the Problem

In this study, we investigated the effects of: (1) the number of PPT measurements; (2) the target muscles (elbow flexor, knee extensor, and ankle plantar flexor muscles); and (3) differences between the sexes in the 20 PPT measurements. Therefore, 20 consecutive PPT measurements were performed on the elbow flexor muscles, knee extensor muscles, and ankle plantar flexor muscles of the dominant side. The target muscles were randomly ordered for each participant. The PPT measurement interval was 15 s between consecutive measurements.

### 2.2. Participants

Thirty healthy adults (15 males and 15 females) participated in the study (male mean ± SD: age, 22.3 ± 0.9 years; height, 170.8 ± 5.8 cm; weight, 67.3 ± 10.8 kg; female mean ± SD: age, 21.9 ± 1.0 years; height, 157.2 ± 4.5 cm; weight, 51.9 ± 7.6 kg). Participants were fully informed about the procedures and aims of the study, after which they provided written informed consent. The study complied with the requirements of the Declaration of Helsinki and was approved by the Ethics Committee of Niigata University of Health and Welfare, Niigata, Japan (Procedure #18615).

The required sample size for a repeated measures two-way analysis of variance (ANOVA) (effect size = 0.10 [medium when considering interaction effects for 2-way ANOVAs], αerror = 0.05, and power = 0.95) using G* power 3.1 software (Heinrich Heine University, Dusseldorf, Germany) was more than 8 participants in each group.

### 2.3. Pain-Pressure Threshold (PPT)

PPT measurements were conducted using an algometer (NEUTONE TAM-22 (BT10); TRY-ALL, Chiba, Japan). With continuously increasing pressure, the soft tissue in the measurement area was compressed with the metal rod of the algometer. The participant was instructed to immediately press a trigger when pain, rather than just pressure, was experienced. The value read from the device at this time point (kilograms per square centimeter) corresponded to the PPT.

For the elbow flexor muscle measurement, the shoulder joint was in mild abduction in the supine position, and measurements were taken at the 70% distal position between the acromion and the lateral epicondyle of the humerus. For the knee extensor muscles, measurements were taken at the 50% position between the superior anterior iliac spine and the bottom of the patella in the supine position. For the ankle plantar flexor muscles, measurements were taken at the medial gastrocnemius muscle belly at the proximal 30% position from the knee crease to the lateral malleolus in the prone position. The specific measurement locations were marked with a pen, which allowed the reproduction of the measurement.

### 2.4. Statistical Analysis

SPSS (version 28.0; IBM Corp., Armonk, NY, USA) was used for the statistical analysis. To verify the consistency of the baseline by sex, we compared the results using an unpaired *t*-test. Repeated measures of two-way ANOVA (gender [male vs. female] × number of measurements [1,2,3,4,5,6,7,8,9,10,11,12,13,14,15,16,17,18,19]) were used to identify the interactions and main effects. Multiple comparison tests with Bonferroni correction for frequency were performed as post hoc tests.

## 3. Results

### 3.1. Comparison of PRE Values among Males and Females

There were no significant differences in baseline assessments at all muscles between males and females.

### 3.2. Elbow Flexor Muscles

No interaction was found in the split-plot analysis of variance, but there was a main effect for the number of measurements (*p* < 0.01). Measurement 10 was significantly different from the 2nd measurement. Measurements 8, 10, 11, and 15 to 10 were significantly different from the 3rd measurement. Measurements 10 and 15 to 20 were significantly different from the 4th measurement. Measurements 18 to 20 were significantly different from the 5th measurement. Measurements 16 and 18 to 20 were significantly different from the 6th measurement. Measurements 16, 18, and 20 were significantly different from the 7th measurement. All post-test results for the measurements are presented in Figure 1.

### 3.3. Knee Extensor Muscles

No interaction was found in the split-plot analysis of variance, but there was a main effect for the number of measurements (*p* < 0.01). Measurements 9, 14, 16, 18, and 20 were significantly different from the 2nd measurement. Measurement 9 was significantly different from the 3rd measurement. Measurements 9, 12, 14, 18, and 20 were significantly different from the 4th measurement. Measurements 9, 12, 14, and 18 were significantly different from the 5th measurement. Measurements 9 and 14 were significantly different from the 6th measurement. All post-test results for the measurements are presented in Figure 2.

### 3.4. Ankle Plantar Flexor Muscles

A significant interaction effect (*p* < 0.05) was observed, and a main effect (*p* < 0.01) was also observed for the number of measurements. The post-test results showed no significant difference in the number of measurements in the male participants. However, the value was significantly (*p* < 0.05) higher in the 14th measurement compared to the 5th measurement in the female participants (Figure 3).

## 4. Discussion

This was the first study to have investigated the effects of repeated PPT assessments (i.e., 20) in the elbow flexor, knee extensor, and ankle plantar flexor muscles on PPT. The main finding was that there was an increase in PPT in the elbow flexors and knee extensors, starting with the eighth and ninth assessments (out of 20), respectively, compared to the second assessment. However, no changes between the first PPT assessment and the other assessments (2–20) were detected. In addition, there was no significant difference in the baseline PPT values of any muscle between the female and male participants. This study showed that repeated PPT assessments can increase the PPT in the elbow flexors and knee extensors, but not in the ankle plantar flexor muscles. Consequently, the induced pain of the PPT assessment itself adjusted the pain sensitivity by an increase in the PPT after several bouts. This has also been reported by various intervention studies which induced a single stimulus (e.g., stretching, foam rolling) until the pain threshold and reported increased PPT after the intervention compared to the baseline values [4,7,8]. Surprisingly, no changes in PPT assessments occurred between the first PPT and all the other remaining assessments (2–19). However, it has to be noted that there was, surprisingly, a decrease in the PPT when just comparing the first and second assessments. Without considering the *p*-value correction of the post hoc tests (i.e., Bonferroni), there was a significant decrease between the first PPT assessment and the second assessment in the ankle plantar flexor muscles (*p* = 0.01) and a tendency of a decrease in the elbow flexor (*p* = 0.07) and knee flexor (*p* = 0.07) muscles. Consequently, it can be assumed that, in the second assessment, the PPT likely increased due to a familiarization effect. This finding could be important for future acute intervention studies which intended to assess the PPT only once before and after an intervention, since the PPT might be underestimated in the post-intervention trial. This could consequently hide a potential acute change in the PPT based on the intervention.

According to our findings, we can recommend for future acute intervention studies and also for clinical applications that between two and a maximum of seven PPT assessments should be applied. Consequently, to obtain reliable data, an average of the first to the seventh PPT assessments is recommended, to avoid overestimating the PPT.

Comparing potential sex differences, a previous study found significantly higher PPTs in males compared to females, in both the rectus femoris muscle and vastus lateralis muscle [9]. Although we also assessed the knee extensor PPT at the rectus femoris muscle, we did not find a significant difference between the baseline PPT of males and females. However, it has to be noted that we did find a tendency (*p* = 0.07) of higher baseline PPT values in the rectus femoris in males (4.11 ± 1.38 kg), compared to females (3.17 ± 1.19 kg). A potential explanation for the insignificant finding might be in the different methodology used in the study of da Silva et al. (2021) [9], compared to the current study. The researchers da Silva et al. (2021) assessed the rectus femoris PPT at the proximal, medial, and distal part of the muscle, while, in our study, we measured the PPT at the mid-portion of the rectus femoris only. Since it is well known that muscle structure, such as stiffness, can differ between muscles [13,14,15], and also within muscles (e.g., proximal compared to distal) [16], future studies should consider measuring the PPT at various locations on the muscle (e.g., proximal, mid-portion, distal).

In general, pain can be considered as a multidimensional phenomenon. It does not only depend on the physiological state since the psychological, social as well as the spiritual state have to be considered. Additionally, receptor pain (e.g., skin, osteo-articular, muscular) and non-receptor pain (e.g., neuropathic) have to be distinguished. Pain receptors in the muscles (i.e., nociceptors) convert electrical impulses and transfer it to centers (e.g., cerebral cortex, limbic system) where pain is recognized. Due to this multidimensional approach, pain should be considered as an individual phenomenon [17]. Besides the visual analogue scale (VAS), the PPT can also measure pain. The latter is assessed with algometers. Algometer measurements can be considered as a cost- and time-efficient alternative for measuring pain sensitivity, compared to other more comprehensive assessments. Various studies have investigated the pain or stretch tolerance of a whole muscle-tendon unit with a dynamometer in a stretched position [18,19]. Hence, the advantage of algometers might be that they can assess pain locally in the various tissues in a rested state. A potential disadvantage compared to a dynamometer assessment is that pain can only be measured at a specific location rather than throughout a whole muscle-tendon unit. This neglects further potential structures such as ligaments, capsules, and fascia, which might also play a role in pain sensitivity.

## 5. Conclusions

We conclude that the PPT increases in the elbow flexors and knee extensors, but not in the ankle plantar flexors, following various repetitions of a PPT assessment. Consequently, for future studies but also for practitioners, it is recommended to use the average of a range of two to a maximum of seven assessments, to not overestimate the PPT.

## Figures and Tables

**Figure 1 healthcare-11-00475-f001:**
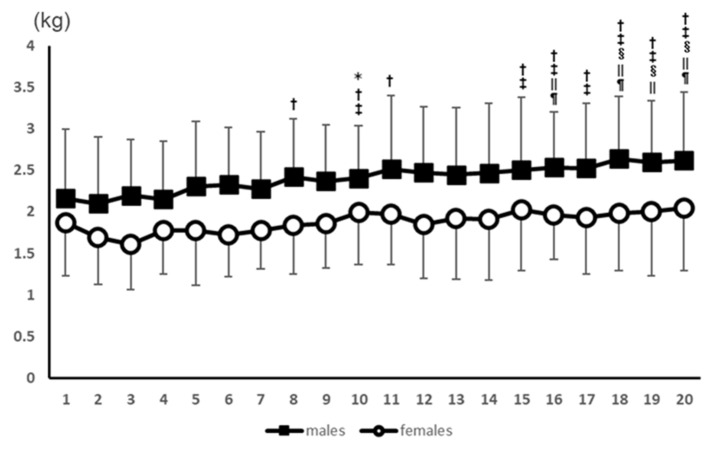
Results of the PPT values for the elbow flexors for 20 consecutive measurements. *: Significant difference from the 2nd measurement. †: Significant difference from the 3rd measurement. ‡: Significant difference from the 4th measurement. §: Significant difference from the 5th measurement. ||: Significant difference from the 6th measurement. ¶: Significant difference from the 7th measurement.

**Figure 2 healthcare-11-00475-f002:**
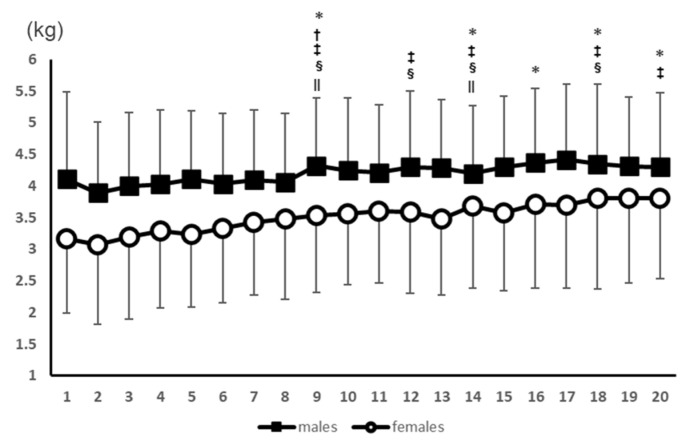
Results of the PPT values for the knee extensors for 20 consecutive measurements. *: Significant difference from the 2nd measurement. †: Significant difference from the 3rd measurement. ‡: Significant difference from the 4th measurement. §: Significant difference from the 5th measurement. ||: Significant difference from the 6th measurement.

**Figure 3 healthcare-11-00475-f003:**
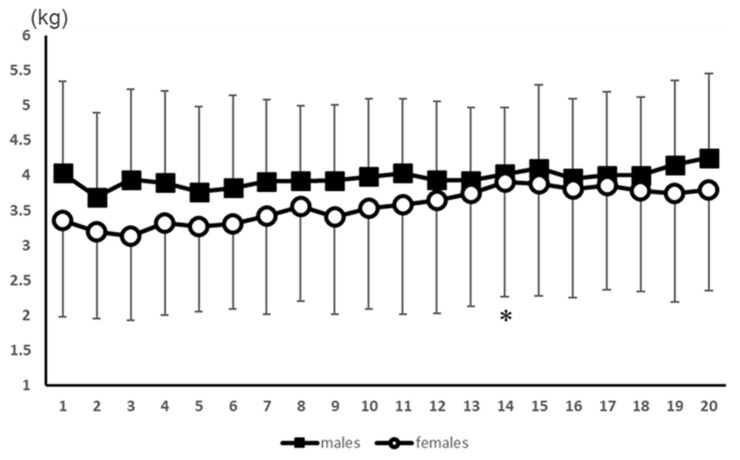
Results of the PPT values for the ankle plantar flexors for 20 consecutive measurements. *: Significant difference from the 5th measurement in females.

## Data Availability

All data supporting the conclusions of this study will be fully provided upon request by the authors.

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
