# Peer review of "Pain-Pressure Threshold Changes throughout Repeated Assessments with No Sex Related Differences"

_healthcare, 2023, doi:10.3390/healthcare11040475_

Round 1
Reviewer 1 Report
COMMENTS
Title: Pain-pressure threshold changes throughout repeated assessments with no sex related differences by Andreas Konrad et al
General: The present manuscript analysed Pain-pressure threshold changes throughout repeated assessments with no sex related differences. The author describe the goal of this study which was perform multiple PPT assessments (i.e., 20) in three different muscles of the dominant side (elbow flexor, knee extensor, and ankle plantar flexor muscles) to detect if there is a shift in PPT throughout the repeated PPT assessments, further goal of the study was to detect the differences between sexes in the potential change of PPT throughout the repeated assessments.
2. Specific
a) Title of the paper is appropriate.
b) What is the basis of taking the PPT?
c) Institutional ethical clearance should be attached
d) Results are poorly presented. Biochemical assays should be added if possible
e) No biochemical and molecular evidences were given to prove the change of PPT behavior
f) As a whole the entire protocol of the studies and write up is a very poor quality in scientific contribution.
g) Discussion is poor, should add more points to address the PPT.
3. Conclusion
This is a poorly represented article the authors are experienced in this area. To my opinion, the article is up to the standard for publication in and may be accepted after the above queries.
Author Response
Reviewer 1:
Title: Pain-pressure threshold changes throughout repeated assessments with no sex related differences by Andreas Konrad et al
General: The present manuscript analysed Pain-pressure threshold changes throughout repeated assessments with no sex related differences. The author describe the goal of this study which was perform multiple PPT assessments (i.e., 20) in three different muscles of the dominant side (elbow flexor, knee extensor, and ankle plantar flexor muscles) to detect if there is a shift in PPT throughout the repeated PPT assessments, further goal of the study was to detect the differences between sexes in the potential change of PPT throughout the repeated assessments.
Thank you for your helpful comments and suggestions that have enabled significant improvements to be made to the manuscript. The authors are also grateful for the reviewer taking the time to evaluate this manuscript. We hope that the revised manuscript satisfactorily addresses all the raised issues. The responses to the reviewer comments are below.
Please note that the page and line numbers in our replies refer to those of the present submission and that all changes in the manuscript are marked with the track changes tool.
- Specific
- a) Title of the paper is appropriate.
Thank you!
- b) What is the basis of taking the PPT?
In our field we use PPT to examine potential mechanism which might explain changes in joint range of motion after a variety of training stimuli such as stretching, foam rolling and massage. The purpose of this specific study was to examine the time course of pain thresholds with repeated application of PPTs, since to date it was not clear if a repeated application of a PPT assessment can adjust the pain thresholds of the various muscles. This is an important information for future studies which apply PPT, to overcome potential methodological flaws.
- c) Institutional ethical clearance should be attached
We have included an ethical statement of this study at the end of the Participants section as the following:
“The study complied with the requirements of the Declaration of Helsinki and was ap-proved by the Ethics Committee of Niigata University of Health and Welfare, Niigata, Japan (Procedure #18615).”
- d) Results are poorly presented. Biochemical assays should be added if possible
We thank for clarifying this. We have now added extended the results section with all the post-hoc tests as the following:
Measurement 10 was significant different to the 2nd measurement. Measurements 8, 10, 11, and 15 to 10 were significant different to the 3rd measurement. Measurements 10 and 15 to 20 were significant different to the 4th measurement. Measurements 18 to 20 were significant different to the 5th measurement. Measurements 16 and 18 to 20 were significant different to the 6th measurement. Measurements 16, 18, and 20 were significant different to the 7th measurement. All post-test results for the measurements are presented in Fig. 1.”
“Measurements 9, 14, 16, 18, and 20 were significant different to the 2nd measurement. Measurement 9 was significant different to the 3rd measurement. Measurements 9, 12, 14, 18, and 20 were significant different to the 4th measurement. Measurements 9, 12, 14, and 18 were significant different to the 5th measurement. Measurements 9 and 14 were significant different to the 6th measurement. All post-test results for the measurements are presented in Fig. 2.”
However, our only parameter assessed of three different muscles in this study was PPT. That’s why we have also decided to go for a short communication. Though, we believe that our findings can be important for further study designs as well as for the practical application.
- e) No biochemical and molecular evidences were given to prove the change of PPT behavior
- f) As a whole the entire protocol of the studies and write up is a very poor quality in scientific contribution.
- g) Discussion is poor, should add more points to address the PPT.
- Conclusion
This is a poorly represented article the authors are experienced in this area. To my opinion, the article is up to the standard for publication in and may be accepted after the above queries.
We are sorry that we have missed all the parts stated by the reviewer. This study aims to understand how many PPT assessments are required to assess pain threshold properly. Therefore, we have assessed PPT of three muscles only, without considering any other explanatory mechanism. That’s why we have submitted this paper as a short communication only (rather than a full article). However, we fully understand the concerns of the reviewer and hence, we have added a paragraph in the discussion section where we discussed the physiology of pain as well as of PPT. We have implemented that in the discussion as the following:
“In general, pain can be considered as a multidimensional phenomenon. It does not only depend on the physiological state since the psychological, social as well as the spiritual state have to be considered. Additionally, receptor pain (e.g., skin, osteo-articular, muscular) and non-receptor pain (e.g., neuropathic) have to be distinguished. Pain receptors in the muscles (i.e., nociceptors) convert electrical impulses and transfer it to centers (e.g., cerebral cortex, limbic system) where pain is recognized. Due to this multidimensional approach pain should be considered as an individual phenomenon. Besides visual analogue scale (VAS) also PPT can measure pain. The latter is assessed with algometers. Algometer measurements can be considered as a cost- and time-efficient alternative for measuring pain sensitivity, compared to other more comprehensive assessments. Various studies have investigated the pain or stretch tolerance of a whole muscle-tendon unit with a dynamometer in a stretched position [15,16]. Hence, the advantage of algometers might be that they can assess pain locally in the various tissues in a rested state. A potential disadvantage compared to a dynamometer assessment is that pain can only be measured at a specific location rather than throughout a whole muscle-tendon unit. This neglects further potential structures such as ligaments, capsules, and fascia, which might also play a role in pain sensitivity.”

Reviewer 2 Report
Abstract:
The Abstract section could be expanded with the major findings, especially pointing on the significance of the research. There is too much sentences with “no significant….”. On this way it is not pointed on the importance of your investigation.
Introduction:
While some studies applied only one PPT assessment, others, however, used multiple PPT assessments. - It is not enough one reference for this sentence.
Materials and Methods
1. Why is your examination based on the healthy volunteers and not on the patients with pain?
2. How did you choose the number of participants? Is it 30 patients enough for the adequate and significant results of the study?
3. How did you calculate the power of the study?
Results
Standard deviations are too high. Are then results appropriate for the discussion?
Discussion
You should have more references and cite another studies with this topic. Expand the discussion.
Author Response
Reviewer 2:
Thank you for your helpful comments and suggestions that have enabled significant improvements to be made to the manuscript. The authors are also grateful for the reviewer taking the time to evaluate this manuscript. We hope that the revised manuscript satisfactorily addresses all the raised issues. The responses to the reviewer comments are below.
Please note that the page and line numbers in our replies refer to those of the present submission and that all changes in the manuscript are marked with the track changes tool.
Abstract:
The Abstract section could be expanded with the major findings, especially pointing on the significance of the research. There is too much sentences with “no significant….”. On this way it is not pointed on the importance of your investigation.
We thank for this important comment. We have now revised the abstract according to the reviewer comment.
Introduction:
While some studies applied only one PPT assessment, others, however, used multiple PPT assessments. - It is not enough one reference for this sentence.
We are sorry for this mistake. We have now included two references at this sentence as the following:
While some studies applied only one PPT assessment [8,11], others, however, used multiple PPT assessments [4,12].
Materials and Methods
- Why is your examination based on the healthy volunteers and not on the patients with pain?
The purpose of this study was to examine the time course of pain thresholds with repeated application of PPTs, since to date it was not clear if a repeated application of a PPT assessment can adjust the pain thresholds of the various muscles. This is an important information for future studies which apply PPT, to overcome potential methodological flaws. Most of the training studies (e.g. stretching, foam rolling) which have assessed PPT in the past are dealing with healthy participants without pain. Hence, we have decided not to recruit patients in pain.
- How did you choose the number of participants? Is it 30 patients enough for the adequate and significant results of the study?
We calculated the samples size via G*power and 30 patients is enough for the adequate and significant results of the study. We added the following description:
The required sample size for a repeated-measures two-way analysis of variance (ANOVA) (effect size = 0.10 [medium when considering interaction effects for 2-way ANOVAs], αerror = 0.05, and power = 0.95) using G* power 3.1 software (Heinrich Heine University, Dusseldorf, Germany) was more than 8 participants in each group.
- How did you calculate the power of the study?
Since we have now included the a priori sample size calculation (please see point 2 of the reviewer) we are confident that we have meet the power of our results as well!
Results
Standard deviations are too high. Are then results appropriate for the discussion?
When considering our results and compare it to previous studies we have achieved similar standard deviations. Whist the standard deviation of our experiments (3 muscles) ranged between 32% to 35% to the respective mean values, other studies showed even higher contribution of standard deviations to mean values (34% to 41%; Jay et al., 2014). Hence, it can be assumed that PPT or pain in general has a very high variation. Consequently, we are confident that our results can be used for discussion.
Jay, K., Sundstrup, E., Søndergaard, S. D., Behm, D., Brandt, M., Særvoll, C. A., ... & Andersen, L. L. (2014). Specific and cross over effects of massage for muscle soreness: randomized controlled trial. International journal of sports physical therapy, 9(1), 82.
Discussion
You should have more references and cite another studies with this topic. Expand the discussion.
To the best knowledge to date no similar study on this topic exists. The lack of previous research and hence, missing studies was also a reason to choose rather a short communication than a full article. However, we have incorporated further points to discuss based on the suggestions of Reviewer 1 as the following:
“In general, pain can be considered as a multidimensional phenomenon. It does not only depend on the physiological state since the psychological, social as well as the spiritual state have to be considered. Additionally, receptor pain (e.g., skin, osteo-articular, muscular) and non-receptor pain (e.g., neuropathic) have to be distinguished. Pain receptors in the muscles (i.e., nociceptors) convert electrical impulses and transfer it to centers (e.g., cerebral cortex, limbic system) where pain is recognized. Due to this multidimensional approach pain should be considered as an individual phenomenon. Besides visual analogue scale (VAS) also PPT can measure pain. The latter is assessed with algometers. Algometer measurements can be considered as a cost- and time-efficient alternative for measuring pain sensitivity, compared to other more comprehensive assessments. Various studies have investigated the pain or stretch tolerance of a whole muscle-tendon unit with a dynamometer in a stretched position [15,16]. Hence, the advantage of algometers might be that they can assess pain locally in the various tissues in a rested state. A potential disadvantage compared to a dynamometer assessment is that pain can only be measured at a specific location rather than throughout a whole muscle-tendon unit. This neglects further potential structures such as ligaments, capsules, and fascia, which might also play a role in pain sensitivity.”

Round 2
Reviewer 2 Report
I suggest to accept this manuscript.